# Reverse Training to Nurse the Reversal Curse

**Olga Golovneva**
FAIR at Meta
olggol@meta.com

**Zeyuan Allen-Zhu**
FAIR at Meta

**Jason Weston**
FAIR at Meta

**Sainbayar Sukhbaatar**
FAIR at Meta
sainbar@meta.com

## Abstract

Large language models (LLMs) have a surprising failure: when trained on "A has a feature B", they do not generalize to "B is a feature of A", which is termed the Reversal Curse. Even when training with trillions of tokens this issue still appears due to Zipf's law – hence even if we train on the entire internet. This work proposes an alternative training scheme, called *reverse training*, whereby all words are used twice, doubling the amount of available tokens. The LLM is trained in both forward and reverse directions by reversing training strings while preserving (i.e., not reversing) chosen substrings, such as entities. We show that data-matched reverse-trained models provide superior performance to standard models on standard tasks, and compute-matched reverse-trained models provide far superior performance on reversal tasks, helping resolve the reversal curse issue.

## 1 Introduction

Large Language Models (LLMs) trained on internet-scale data perform extremely well on tasks relating to reasoning, common-sense, and world-knowledge. In particular, the range of knowledge captured by LLMs like GPT-4 (OpenAI, 2023) and Llama-2 (Touvron et al., 2023b) is significantly wider than that of an average person. However, recent research (Berglund et al., 2023b; Allen-Zhu & Li, 2023a;b) uncovered a curious flaw in the knowledge capabilities of LLMs, coined the *reversal curse*. They experimentally showed that even the currently most powerful LLMs are incapable of "reversing" facts they had learned. For example, standard LLMs cannot correctly answer "What's the capital of France?" even if the training data contains "Paris is the capital of France", unless it also contains text where "France" is followed by "Paris", such as "As the capital of France, Paris has ...".

This is a serious problem because it means LLMs cannot learn the equivalence of relations like "A is the capital of B" equals "B's capital is A" despite being trained on many pairs of such facts. In contrast, a human child can learn such general rules from just a few observations of both directions, which makes it an elementary function of human intelligence. The reversal curse may have been hard to notice at first because most LLMs are trained on internet-scale data, which is likely to contain the most common facts in both directions. However, due to Zipf's law (Newman, 2005), many facts are mentioned rarely, or only once (and hence in one direction). Further, a more common concept can still be attached to more rare concepts, for example the names or details of a celebrity's parents. Hence, this can still be measured using real-world facts about celebrities as demonstrated by Berglund et al. (2023b). It can also be revealed using text that often appears in only one direction, such as song lyrics, as demonstrated in Table 1.

In this paper, we propose a simple training method to reduce the effect of the reversal curse. We first observe that LLMs are trained in an autoregressive way from left-to-right, which may contribute to the reversal curse. While predicting the next word might be more natural, it is also possible to train a LLM in the right-to-left direction, by predicting the previous word from its subsequent words. Such reverse training has the potential to solve the reversal curse because it allows the model to see a fact in its reverse direction. However, this knowledge has to be transferred to test time left-to-right generations. Viewing the reversed text as a second language, it is known that training on multiple varying sources can be leveraged to help each other via multitasking (e.g., from code to math (Shao et al., 2024),

| **Query:** What is the line that comes after "Gave proof through the night that our flag was still there" in the US anthem? | **Query:** What is the line that comes before "O say does that star-spangled banner yet wave" in the US anthem? |
|---|---|
| **GPT4**[1]**:** The line that comes after "Gave proof through the night that our flag was still there" in the U.S. national anthem, "The Star-Spangled Banner," is: "O say does that star-spangled banner yet wave" | **GPT4:** The line that comes before "O say does that star-spangled banner yet wave" in the US National Anthem, "The Star-Spangled Banner," is "And the rocket's red glare, the bombs bursting in air." |
| **Llama-2 Chat:** The line that comes after "Gave proof through the night that our flag was still there" in the US anthem is: "O say does that star-spangled banner yet wave ... | **Llama-2 Chat:** The line that comes before "O say does that star-spangled banner yet wave" in the US anthem is: "O long may it wave o'er the land of the free and the home of the brave." |

Table 1: **An example of the reversal curse:** capable LLMs fail to recall a fact in reverse if it is mostly seen only in one direction, such as lines of song lyrics. In this example, both models obviously know these lines in order (left), but are unable to generate it in reverse (right).

or in cross-lingual pre-training (Lample & Conneau, 2019)). We investigate four reversal types that require minimal processing and can be seen as this additional "language": token reversal, word reversal, entity-preserving reversal and random segment reversal. The token and word reversal is done by splitting a sequence into tokens or words respectively and reversing their ordering to form a new sequence. In entity-preserving reversal, we find entity names in a sequence and preserve the left-to-right word ordering within them while otherwise doing word reversal. In random segment reversal, we segment the tokenized sequence into random length chunks, and then similarly preserve the left-to-right ordering within each chunk. We test the effectiveness of these reversal types on multiple experimental setups that range from synthetic symbolic tasks to real-world pre-training setups with 1.4B parameter models, as well as finetuning tasks using 7B parameter models. Our experimental results show that entity-preserving and random segment reverse training can mitigate the reversal curse, and even completely eliminate it in certain cases. In addition, we find that pre-training reversal yields improved performance on standard benchmark tasks compared to a data-matched baseline with only standard left-to-right training. Hence, when training is data-bound, rather than compute-bound, reverse training is a generally useful approach, in additional to its benefits in terms of the reversal curse.

## 2 Reverse Training

Reverse training consists of taking a training dataset with $N$ samples $\{x_1, \ldots, x_N\}$ and constructing the set of reversed samples

$$\overleftarrow{x_i} = \text{REVERSE}(x_i), \quad i = 1, \ldots N.$$

Training is then conducted using the combined set $\{x_i\} \cup \{\overleftarrow{x_i}\}$ of $2N$ training samples, using the typical language modeling objective. The function $\text{REVERSE}(\cdot)$ reverses the given string, where we consider various choices of reversal type:

- **Token reversal** ($\text{REVERSE}_{token}$): A given input $x_i$, when tokenized, e.g. using BPE (Sennrich et al., 2015), consists of tokens $x_i^t$, and the reversed version has the form $\overleftarrow{x_i}^t = x_i^{|x_i|-t+1}$.

- **Word reversal** ($\text{REVERSE}_{word}$): Each example is first split into words.[2] We then reverse the string at the word level, joining it back together with spaces. Note that this input would then typically be tokenized for input into the LLM, e.g. using BPE.

---

[1]This example used the GPT4 model accessed at `https://chat.openai.com/` on Mar 4th, 2024.
[2]We use the word splitter in NLTK (Loper & Bird, 2002).

| Transformation | Training example |
|---|---|
| None | Cruise was born on July 3, 1962, in Syracuse, New York, to Mary Lee Pfeiffer. |
| Word reversal | . Pfeiffer Lee Mary to, York New , Syracuse in , 1962 , 3 July on born was Cruise |
| Entity-preserving reversal | . Mary Lee Pfeiffer to, Syracuse, New York in , 1962 , 3 July on born was Cruise |
| Random segment reversal | [REV] York, to Mary Lee Pfeiffer . [REV] in Syracuse, New [REV] on July 3, 1962,  [REV] born [REV] Cruise was |

Table 2: **Reversal transformations:** examples of different reversal types on a given string. In practice, training examples can be much longer (e.g., entire documents during pre-training). The language model is still trained left-to-right on such transformations, and in the word reversal case is essentially predicting the sentence backwards (right-to-left) starting from the last word. The entities which have their word ordering preserved are highlighted by underlines. In random segment reversal, the segments are separated by "[REV]". *Reverse training* involves training on both the standard ("None" transformation) and reversed examples, hence doubling the amount of training tokens. The reverse transformation can be seen as a second "language" the model has to learn, note this is not the same as reversing the relation between facts, which remains intact, as the model can tell from the syntax whether it is in forward or reverse language prediction mode.

- **Entity-preserving reversal** (REVERSE$_{entity}$): We run an entity detector over a given training sample[3], which also splits the non-entities into words. We then reverse the words, but keep the word-order of entities in their original left-to-right order. The string is then joined as before with spaces. See Table 2 for an example.

- **Random segment reversal** (REVERSE$_{rand}$): Instead of running a relatively costly segmentation such as an entity detector, we experiment with randomly segmenting the sequence into chunks of size between 1 and $k$ tokens using uniform sampling. We then reverse the segments, but keep the word order within each segment in their original left-to-right order. The segments are then joined with a special token "[REV]", which indicates the end of left-to-right prediction for the given segment. During training epochs, each time the example is seen we perform a different random segmentation to increase diversity. See Table 2 (last row) for an example.

Both forward and reversed training samples are shuffled together so that training batches can contain random (unpaired) examples of both types. In our experiments, we perform reverse training at both the pre-training and finetuning stages, but also ablate these variants to analyze their impact.

One can view the extra data $\{\overleftarrow{x_i}\}$ as another language that the language model has to learn left-to-right – in this case a reversed natural language, which has a similar difficulty in terms of perplexity. As it is easy for the language model to identify which of these languages it is trying to generate from when predicting the next token, this does not tend to interfere with its language modeling abilities in the standard forward direction. Further, as it has been shown that LLMs can leverage knowledge across different sources (e.g., code to math (Shao et al., 2024), or different natural languages (Lample & Conneau, 2019)) we hypothesize that the knowledge it learns from the reverse direction can help in the forward direction as well.

Another perspective of reverse training is from an information theory viewpoint. The language modeling objective is to learn the probability distribution of natural language, which can be conveniently decomposed into next token predictions for each sample $x_i$

$$p(x_i^1, \ldots, x_i^{|x_i|}) = \prod_{t=1}^{|x_i|} p(x_i^t | x_i^1, \ldots, x_i^{t-1}).$$

---

[3]We use the flair/ner-english-large model for entity detection (Schweter & Akbik, 2020).

| Training method | Entity name length | | |
|---|---|---|---|
| | 2 words | 3 words | 5 words |
| standard | 0.0 | 0.0 | 0 |
| reverse training (*word*) | 95.8 | 16.9 | 2.0 |
| reverse training (*entity*) | 100.0 | 100.0 | 100.0 |
| reverse training (*rand k=2*) | 100.0 | 98.4 | 22.7 |
| reverse training (*rand k=3*) | 100.0 | 100.0 | 79.2 |
| reverse training (*rand k=5*) | 100.0 | 100.0 | 100.0 |

Table 3: Test accuracy (%) on the *symbolic reverse* task. Standard training completely fails. Word reversal works well for shorter entities, but entity preserving reversal is necessary for entities with more words. Random segment reversal performs well when the maximum segment length $k$ is at least as long as the entities.

While this left-to-right direction is more natural, the same probability can be decomposed in the reverse direction as well

$$p(x_i^1, \ldots, x_i^{|x_i|}) = \prod_{t=|x_i|}^{1} p(x_i^t | x_i^{t+1}, \ldots, x_i^{|x_i|}).$$

If we make an assumption that LLM's language capabilities are partially due to learning to compress natural language (Del'etang et al., 2023) according to the source coding theorem (Shannon, 1948), then training in the reverse direction towards the same perplexity should also acquire some of those capabilities. For example, filling the blank in "__ is the capital of France." requires a similar level of language understanding and world knowledge as predicting the next word of "Paris is the capital of __".

## 3 Experiments

### 3.1 Symbolic reverse task

We first create a simple symbol-based (rather than natural language-based) toy dataset to investigate the reversal curse in a controlled setting. We start by randomly pairing entities $a_i$ and $b_j$ in a one-to-one manner. The training data contains all the forward $a_i \to b_j$ mappings, but only half of the backward $b_j \to a_i$ mappings. The remaining backward mappings form the test data. To succeed in this task, a model must infer the rule $a_i \to b_j \Leftrightarrow b_j \to a_i$ from the training data, then generalize it to the pairs in the test data.

The entity names are created by combining multiple random code words, e.g. two-word entities $a_i$ = "a12 a64" and $b_j$ = "b54 b42". Each sample contains a mapping written like "a12 a64 has a feature b54 b42" or its reverse "b54 b42 is a feature of a12 a64". We use the simple word tokenization, which makes REVERSE$_{token}$ equivalent to REVERSE$_{word}$. For evaluations, we report the exact match accuracy of the target entity (2nd entity), averaged over three random seeds. More training details are given in Appendix A.

Table 3 shows the performance of reverse training on this task for different entity name lengths. The standard language model training completely fails despite the simplicity of this task, suggesting that it is unlikely to be solved by scaling alone. In contrast, REVERSE$_{word}$ training nearly solves it for two-word entities, but its performance degrades quickly as the entities become longer. A possible explanation is that while REVERSE$_{word}$ does see both mapping $a_i \to b_j$ and its reverse $\overleftarrow{b}_j \to \overleftarrow{a}_i$, it struggles with converting between entity $a_i$="a12 a64 a22" and its reverse $\overleftarrow{a_i}$ ="a22 a64 a12" when they have more words. Matching this reversed entity itself looks similar to the problem in the original reversal curse, which we know is hard to learn. REVERSE$_{entity}$ training eliminates this issue because the word ordering within entity $a_i$ remains the same, which explains its perfect accuracy even for 5-word entities.

| Pre-training method | full name recall (%) | | | | last name recall (%) | | | |
|---|---|---|---|---|---|---|---|---|
| | all | f=4 | f=3 | bdate | all | f=4 | f=3 | bdate |
| *bioS* | | | | | | | | |
| standard | 0.0 | 0.0 | 0.0 | 0.0 | 0.2 | 0.1 | 0.2 | 0.1 |
| reverse training (*token*) | 0.0 | 0.0 | 0.0 | 0.0 | 63.7 | 62.8 | 48.1 | **0.2** |
| reverse training (*word*) | 0.0 | 0.0 | 0.0 | 0.0 | **99.3** | **99.0** | **91.1** | 0.1 |
| reverse training (*entity*) | 99.0 | **98.8** | **87.8** | 0.0 | 0.3 | 0.3 | 0.3 | **0.2** |
| reverse training (*rand k = 25*) | **99.8** | 98.5 | 80.7 | 0.0 | 1.1 | 1.0 | 0.6 | **0.2** |
| *bioR* | | | | | | | | |
| standard | 0.0 | 0.0 | 0.0 | 0.0 | 0.2 | 0.1 | 0.1 | 0.1 |
| reverse training (*token*) | 0.0 | 0.0 | 0.0 | 0.0 | 60.9 | 58.5 | 49.1 | **0.2** |
| reverse training (*word*) | 0.1 | 0.0 | 0.0 | 0.0 | **99.2** | **98.4** | **94.4** | **0.2** |
| reverse training (*entity*) | 97.8 | 92.2 | 78.7 | **0.1** | 0.4 | 0.4 | 0.3 | **0.2** |
| reverse training (*rand k = 25*) | **98.9** | **97.8** | **94.9** | **0.1** | 8.6 | 8.4 | 7.0 | **0.2** |

Table 4: Evaluation results on the *reversing biography* tasks in the mixed-training setup (see Footnote 4, pre-train+FT is deferred to Appendix C). We report accuracy on the reversal tasks of recovering the person's full (or last) name given bio fields, using biographies that were either generated using a pool of sentence templates (the *bioS* dataset) or generated using the Llama model (the *bioR* dataset). We consider when all 6 or f= 3, 4 selected bio fields are given, as well as when only birthdates are given.

REVERSE$_{rand}$ can also solve this task, but only when the maximum segment length $k$ is long enough. When $k$ is smaller than the entity name length, the entity names will always split across multiple segments, thus the same issue as with word reversal could arise.

## 3.2 Reversing biography task

When the reversal curse was discovered in Allen-Zhu & Li (2023b), the authors utilized a biography dataset of 100K randomly generated individuals with unique English names. The biographies were either generated using a pool of sentence templates (the bioS dataset) or generated using the Llama (Touvron et al., 2023a) model (the bioR dataset). The biography entries always start with the person's full name. The reversal QA task is, therefore, very natural: given a person's partial or full biography details, ask for the person's name.[4]

We conducted the same experiment in their setting, with respect to token, word, entity-preserving, and random segment reversals. Our main findings can be summarized as follows (see Table 4, and Appendix C Table 10):

- For the reversal tasks of determining the person's full name, only in the entity-preserving or random segment reversal cases do accuracies become non-trivial. Both token/word reversals completely fail in such tasks.

  - When determining a person's name given only their birth date, the accuracy of reversal tasks remains near zero. This aligns with the "reverse6" task results in Allen-Zhu & Li (2023b): after reversals, the person's name appears near the end of the biography, so that the model stores the person's name *jointly into* all their attributes. Therefore, providing only one attribute is insufficient for the model to accurately identify the person's name.[5]

- If the reversal tasks are simplified to determining the person's *last name only*, then word-level reversal suffices, and token-level reversal also yields non-trivial accuracies.

---

[4]Authors consider two types of training, pre-train + finetune (FT) in which they pre-train the model with biography entries and then finetune with QA tasks; mixed-training in which they train the model once with both the biography entries and the QA tasks (not in the same context). They always use half of the individuals' QA tasks for training and evaluate the QA accuracies on the remaining half.

[5]This is also confirmed by the P-probing results in Allen-Zhu & Li (2023a), which demonstrate that knowledge of the last-appearing entity is stored in a complex manner jointly into all prior entities.

| Pre-training method | celebrity → parent | | | parent → celebrity | | |
|---|---|---|---|---|---|---|
| | best@1 | @5 | @10 | best@1 | @5 | @10 |
| *Model size: 1.4B* | | | | | | |
| standard (compute-matched) | **1.6** | 2.9 | 3.9 | 0.9 | 2.9 | 3.9 |
| standard (data-matched) | 0.4 | 1.7 | 2.7 | 0.8 | 1.8 | 3.2 |
| reverse training (*token*) | 0.8 | 2.5 | 3.8 | 0.6 | 2.5 | 3.9 |
| reverse training (*entity**) | 0.8 | 2.6 | 3.8 | **3.6** | **8.1** | **10.4** |
| reverse training (*rand k=25*) | 1.2 | **3.1** | **4.5** | 1.6 | 4.1 | 6.6 |

Table 5: Evaluation results on the *real-world celebrity* task when using different pre-training methods with no finetuning. Results are reported as best accuracy when sampling multiple times. Reverse (*entity**) pre-training (5% of the reversed data being entity-preserving reversal, and the rest word-reversal) significantly improves the more challenging parent to celebrity direction. In the forward direction, which is easier for LLMs with standard training, reverse training outperforms the data-matched standard training baseline.

- – Some readers may find it surprising that an entity-preserving or random segment method can determine the person's full name but not much the person's last name. This is a known phenomenon (Allen-Zhu & Li, 2023b, partial retrieval): a language model may completely fail at retrieving *later* tokens of a knowledge piece (such as the last name) without the help of spelling out earlier tokens — and the earlier tokens serve as a Chain of Thought (CoT).
- All the reversal methods do not impair the model's performance in forward tasks (such as determining the person's birth dates from names) as shown in Table 10.
- Mixed-training (i.e., adding instruction tuning data to the pre-training level) generally performs better compared to first pre-training the model with knowledge and then fine-tuning it to answer (reversal) tasks. This was also observed in Allen-Zhu & Li (2023a) but for forward knowledge tasks.

More details of this experiment are included in Appendix C.

### 3.3 Reversing real-world knowledge via pre-training

Next we test our method on a realistic setup where we pre-train language models, and evaluate their ability on "forward" and "reverse" facts about real-world knowledge. As LLMs acquire the majority of their world knowledge during their pre-training stage, it makes sense to evaluate our reverse training in this pre-training setup. To make the experiments tractable, we train a Llama-2 1.4 billion parameter model (Touvron et al., 2023b).

We train the baseline model on 2 trillion tokens in the left-to-right direction. Reverse training uses only half of these tokens (1 trillion), but trains in both the standard left-to-right direction, and in the right-to-left (reverse) direction with this same subset of the data. Hence it does model updates over 2 trillion tokens in total, i.e. 1 trillion tokens in each direction is passed through the model. We call this setup *compute-matched* because both models processed the same amount of tokens in total and used the same compute resources. We also train a *data-matched* baseline that is trained on 1 trillion tokens in the standard left-to-right direction. This model has been trained with half as many updates, but has seen the same data examples as the reverse trained model, but only in one direction. We compare these with our reversal approaches: token, entity and random reversal. For entity-preserved reverse training, we employ entity-preserving reversal for 5% of the pre-train data, and the remainder uses word-reversal, mainly due to the extra computational cost of the entity reversal procedure, which we refer to as "entity*" in our results.

To test the reversal capability on real-world facts we use a celebrity task, which contains questions like "The mother of [celebrity_name] is __" that are known to be challenging to large scale LLMs. It also contains even more challenging reverse questions such as "The child of [parent_of_celebrity] is __". We perform two-shot evaluation using our pre-trained models without any finetuning on this dataset.

| Pre-training method | Finetuning method | NameToDescription | | DescriptionToName | |
|---|---|---|---|---|---|
| | | forward | reverse | forward | reverse |
| *Model size: 1.4B* | | | | | |
| standard (compute-matched) | standard | 77.3 | 0.0 | 98.3 | 2.3 |
| standard (compute-matched) | reverse (*entity*) | 78.3 | 85.0 | 99.0 | 5.7 |
| standard (compute-matched) | reverse (*rand k=25*) | 77.3 | 96.3 | 97.7 | 70.7 |
| standard (data-matched) | standard | 75.0 | 0.0 | **99.3** | 0.0 |
| standard (data-matched) | reverse (*entity*) | 75.0 | 66.7 | **99.3** | 3.3 |
| standard (data-matched) | reverse (*rand k=25*) | 76.3 | 94.3 | 95.7 | 67.0 |
| reverse training (*entity**) | reverse (*entity*) | 77.0 | 78.3 | 95.3 | 2.3 |
| reverse (*rand k=25*) | reverse (*rand k=25*) | **81.0** | **97.0** | 97.3 | **73.0** |
| *Model size: 7B* | | | | | |
| standard | standard | **80.3** | 0.0 | 96.0 | 4.0 |
| standard | reverse (*entity*) | 79.0 | 89.7 | **99.7** | 6.0 |
| standard | reverse (*rand k=25*) | 78.3 | **99.0** | 99.0 | **70.0** |

Table 6: Test accuracy (%) on the *fictitious celebrities* task, with either standard (data or compute-matched ) pre-training, or reverse pre-training, and either standard or reverse finetuning, for 1.4B and 7B parameter models. In all cases, reverse finetuning brings a significant improvement on the reverse NameToDescription task, which is otherwise impossible to solve, and to reverse DescriptionToName using random segment reversal.

The results are shown in Table 5. We sample multiple times from the models for each question and if any one of them contains the correct answer, then it is counted as success. The accuracy is relatively low in general due to the small model size in terms of number of parameters, limited pre-training and lack of any finetuning for both baselines and our method. Nevertheless, in the forward direction questions, the reverse training outperforms the data-matched baseline, showing that in the data-bound case, reverse training even helps on standard tasks. The random reversal approach outperforms even the compute-matched case on the direct task for 5 and 10 samples, even though the baseline has effectively access to more data. Importantly, in both the data-matched and compute-matched case we see significant improvement in the reverse direction questions for reverse training compared to either baseline. This demonstrates that reverse training can be employed during the pre-training stage to make the model robust against the reversal curse.

## 3.4 Reversing fictitious facts via finetuning

We next explore if our reverse training can be applied to the finetuning stage when the model is learning new, previously unseen knowledge from a small training dataset. We use the same pre-trained models described in Section 3.3 and an additional Llama-2 7B model, and further finetune them on a dataset made up of fictitious facts. These data are made up of statements of the form "[name] is [description]" (or the reverse) where the names and descriptions are randomly generated. The fictitious nature of this dataset guarantees that those facts are not seen during pre-training, and are only seen in the specified direction during finetuning. The model is then tested to see if it is able to learn these facts in the same or reverse direction that it has seen during training.

Table 6 provides evaluation results for different pre-training and finetuning setups. We employ a soft matching score as the test accuracy, which we evaluate as exact presence of the target sequence in the first 64 tokens of a model's prediction. Across all the pre-trained models, finetuning with reverse training was critical in solving the reversal of NameToDescription, reaching close to 100% for the larger 7B model, while standard finetuning always results in 0% accuracy. For reversing DescriptionToName, only finetuning with the random segment reversal succeeded, achieving an accuracy around 70%. This is likely because generating descriptions is more challenging as they have many words and some variety even for the same person. We observe improvement from reverse pre-training in the data-matched case, but not in the compute-matched case. We note that this is perhaps to be expected as the evaluation statements are fictitious and never appeared in the pre-training data.

| Pre-training method | BoolQ | PIQA | SIQA | HellaS | WinoG | ARCe | ARCc | OBQA | MMLU | Avg |
|---|---|---|---|---|---|---|---|---|---|---|
| *Model size: 1.4B* | | | | | | | | | | |
| std. (compute-matched) | 65.1 | 74.4 | 41.2 | 47.7 | 62.7 | 67.6 | 32.1 | 27.0 | 27.1 | 49.4 |
| std. (data-matched) | 60.5 | 71.6 | 41.5 | 44.5 | 59.9 | 64.2 | 30.0 | 27.2 | 27.9 | 47.5 |
| reverse (*token*) | 63.7 | 72.9 | 41.6 | 45.1 | 60.0 | 65.7 | 30.5 | 28.0 | 25.8 | 48.1 |
| reverse (*entity**) | 62.7 | 72.3 | 40.9 | 45.5 | 59.4 | 65.1 | 29.4 | 25.4 | 27.7 | 47.6 |
| reverse (*rand k=25*) | 63.0 | 73.2 | 41.6 | 46.5 | 62.0 | 67.6 | 31.7 | 26.4 | 27.4 | 48.8 |
| *Model size: 7B* | | | | | | | | | | |
| standard | 77.4 | 78.8 | 48.3 | 77.2 | 69.2 | 75.2 | 45.9 | 58.6 | 45.3 | 64.0 |

Table 7: Performance on standard benchmarks. Reverse training can outperform standard training in the data-matched case, but is behind compute-matched training which uses more data. Llama-2 7B accuracy is provided for reference and is taken from Touvron et al. (2023b).

### 3.5 Analysis & ablation experiments

**Does reversal training hurt performance on standard tasks?**   In Sections 3.1 to 3.4 we showed that reverse training helps to mitigate the reversal curse. Here, we explore if our method disrupts zero-shot performance on common evaluation tasks: BoolQ (Clark et al., 2019), PIQA (Bisk et al., 2020), SIQA (Sap et al., 2019), HellaSwag (Zellers et al., 2019), WinoGrande (Sakaguchi et al., 2021), ARC easy and challenge (Clark et al., 2018), OpenBookQA (Mihaylov et al., 2018). We also report 5-shot performance on the aggregated MMLU benchmark (Hendrycks et al., 2020). Evaluation results are summarized in Table 7. We observe that our random reversal model is 1.3 points better than the standard data-matched 1.4B model trained in the standard forward direction, and is only 0.6 points behind the compute-matched model in accuracy on average, despite being trained on half of the tokens. We note that token reversal works slightly better than entity reversal on these standard benchmarks, and both are superior to the data-matched standard training as well.

We also find that reversed models can not only generate text in the normal left-to-right direction, but can generate the beginning of the text given a continuation — a capability that standard models lack (see an example in Appendix B Table 8). This reverse generation function can be useful in itself, for example for instruction backtranslation (Li et al., 2023).

**Does the unit of reversal matter?**   To understand the effect of segment granularity when reversing sequences, we evaluate the performance of the following training methods on the fictitious celebrities task: standard finetuning, token and word reversal finetuning, entity-preserving reversal finetuning, and random segment reversal finetuning with varying $k$ as described in Section 2. The results are summarized in Appendix B Table 9. In general, we find that reversing at a fine-grained level such as token or word level does not significantly help to resolve the reversal curse, and only improves performance on the reverse tasks by 2-3%. Preserving entities during reversal makes it possible to predict names, but not descriptions. This indicates a close relation between the unit of reversal training and the target "concepts" (e.g. names, descriptions) of the reversal task. Similarly, the random segment reversal performs poorly at predicting descriptions when the segment length limit is set lower than the typical length of a description. The results from Section 3.1 also support this hypothesis.

## 4   Related Work

**Reversal Curse & Mitigations**   The reversal curse was identified by the concurrent works Berglund et al. (2023b); Allen-Zhu & Li (2023b); its name was derived from the former. They demonstrated that the reversal curse occurs across model sizes and families, including very large models such as GPT-3.5 and GPT-4. They found that including auxiliary examples with both orders present in the finetuning or pre-training datasets (to promote meta-learning) does not aid generalization to examples where only one order is given, even if such data is rewritten in a question-answer format. Furthermore, including multiple paraphrases of

each fact in a single direction does not facilitate learning in the reverse direction, despite aiding the given direction, as shown by Berglund et al. (2023a); Allen-Zhu & Li (2023a).

The concurrent work by Allen-Zhu & Li (2023a) investigates a related set of failures and potential solutions. Exploring the capability to answer questions based on synthetic biographies, they examine several data augmentation strategies, including incorporating instruction tuning data into pre-training, generating multiple unique biography entries, permuting biography sentences, and substituting pronouns or partial names with full names. They discover that augmentation during the pre-training phase is essential for enhancing downstream question answering performance across various tasks. However, in real pre-training data, some augmentations may not be feasible — for instance, permuting sentences could degrade language model quality, and it remains uncertain how to best rewrite data during augmentation. Reverse training addresses this issue by presenting a distinct language task (the reversed language) to the language model, thereby avoiding interference with the primary task of left-to-right natural language modeling.

**Right-to-left, masked & other training variants**  Multiple works have proposed pre-training language models with rephrased or paraphrased text (Lewis et al., 2020; Maini et al., 2024), and training right-to-left has been explored before (Pfau et al., 2023; Nguyen et al., 2024), but these works were not targeting the reversal curse. Rather than training left-to-right, or right-to-left, masked language models aim to learn how to "fill in the middle", going back to early language modeling work such as Collobert et al. (2011), and models such as BERT (Devlin et al., 2018). Other methods have also been proposed to explicitly fill in middle text sections by rearranging data (Bavarian et al., 2022), to train on scrambled data (Sinha et al., 2021), train on all permutations of the factorization order (Yang et al., 2019). Relatedly, transforming training data with repeating segments has also been shown to improve language model embeddings (Springer et al., 2024). Encoder-only models akin to BERT have been shown to *not* mitigate the reversal curse (Allen-Zhu & Li, 2023a). However, modifying the architecture and training procedure *has* been shown to help, e.g. by introducing BIdirectional Casual language modeling Optimization (BICO) (Lv et al., 2023). In contrast, our work seeks to rectify the issue while keeping standard language model training as similar as possible to the current regime.

The most similar work to ours is the concurrent work of Guo et al. (2024). They employ various augmentations at the finetuning, rather than pre-training stage, including shuffling and reversing chunks of the input sentences. Unlike our method, their method first segments sentences in the training into semantically meaningful chunks via an LLM. While a chunk can be an entity name, it is more generally applied to all words, e.g. "of developing the first emotional" as a chunk. The actual segmentation is done via prompting another LLM with a specific instruction. Therefore, the unit of reversal will depend on the LLM and its prompt, making it presumably a difficult language modeling problem, whilst also requiring extra compute to reverse the sequence. This is applied only to finetuning on short sentences, which means the reversal curse mitigation is limited to the facts included in the finetuning data, and it is unclear if it can be applied to large pre-training documents. In contrast, our method is applied in the pre-training stage so it can learn to reverse a wide-range of general knowledge facts.

## 5   Conclusion

In this paper, we introduced a simple yet effective training method to help remedy the reversal curse in LLMs. Our reverse training works by first segmenting the input sequence into chunks and then reversing the ordering of chunks, but leaves the word-ordering in each chunk intact. A chunk can be a token, a word, an entity name, or a random number of tokens. The model is then trained on both the original sequences, and this reversed data. We evaluated on a symbolic reverse task and a reversing biography task that both demonstrated the necessity of preserving word-ordering within chunks. Next, we applied our reverse training to the realistic setting of LLM pre-training, which minimized the reversal curse on real-world knowledge. Evaluations on common benchmark tasks reveal that reverse training (particularly random segment reversal) during pre-training does not interfere with

the forward prediction ability of LLMs, and actually improves metrics in the data-bound (rather than compute-bound) setting compared to standard training. When our method is applied to finetuning on fictitious facts, prediction accuracy rose from 0% to 70-100%. The reversal curse is a serious flaw in how LLMs acquire knowledge and reverse training opens a new promising direction in resolving it. While our experiments are focused on English, we expect our method to generalize to other languages, especially the random-segment reversal version because it has less dependence on language.

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

# A  Symbolic reverse task details

The vocabulary is built by generating 100 different words per position, e.g. `a200` to `a299` for the second word of entities $a_i$. Then we create 10,000 entities $a_i$ and 10,000 entities $b_j$ by concatenating random words specific to each position. Finally, $a_i$ entities are randomly mapped to $b_j$, resulting in 10,000 pairs. The model is an 8-layer Transformer with a hidden size of 512. The training is continued for 500 epochs with batch size = 1024, learning rate = 0.0003, and dropout rate = 0.1.

# B  Transformer model pre-training

We train a transformer model with $dim = 2048$, $n\_layers = 24$, and $n\_heads = 16$, resulting in 1.4B parameters. Training data and hyperparameter setup mostly repeats the one from Touvron et al. (2023b). To adapt for the relatively smaller model size, we increase the learning rate to $4.0e{-}4$, and the global batch size was capped at 2M due to the limited number of GPUs. During training, we observe a fixed gap in training perplexity between baseline models and reverse training (Figure 1). The loss of the baseline model is measured on data in the standard direction, while the reverse training loss covers data in both directions. We posit that the reverse training doesn't interfere with forward learning — thus, the model's performance does not degrade on standard benchmarks in data-match conditions, and because we observe a match in the convergence rate of the reverse trained models with the baseline model when it's trained on about 50% of the data.

In Figure 2, we evaluate performance on the real-world knowledge task for multiple checkpoints during pre-training, where accuracy is reported using best@1 sampling. We notice an upward trend in performance on the reverse task with no saturation at the last checkpoint. Hence, we assume that if we continue pre-training we would see further improvement.

We also find that reversed models can not only generate text continuations in the normal left-to-right direction, but can generate the beginning of the text given a continuation — a capability that standard models lack. We give an example in Table 8.

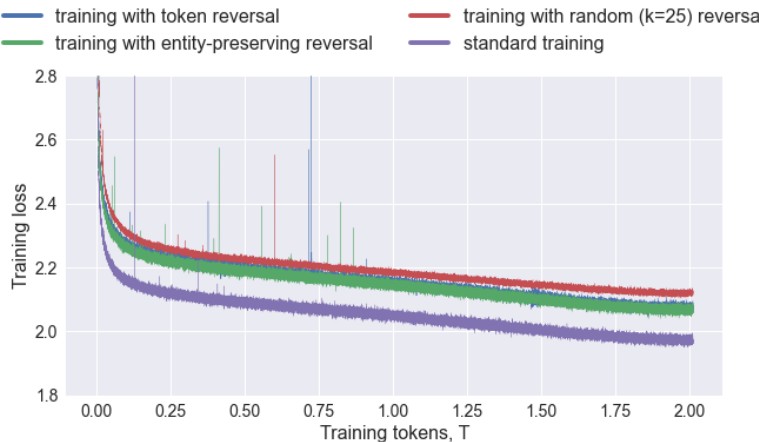

Figure 1: Training loss for 1.4B models in the pre-training stage. On the $x$-axis we display the total number of tokens model has been trained on, including both in standard and reverse direction.

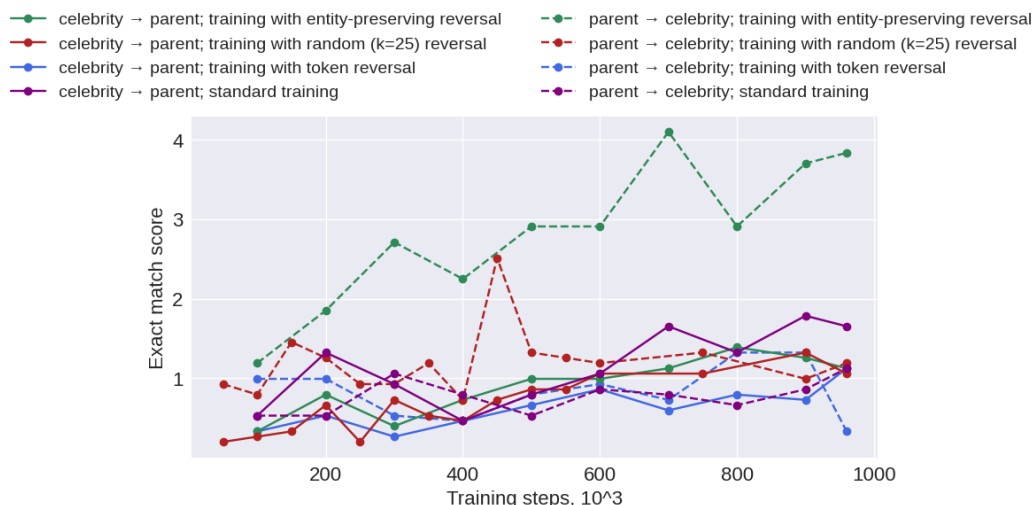

Figure 2: Evaluation results during training on the real-world celebrity task when using different pre-training methods for LLMs.

Finally, we evaluate the performance on the fictitious celebrities task in different ablation setups, varying pre-training and finetuning training approaches. Results are summarized in Table 9.

## C   Biography Data Experiment Details

In our biography data experiments, we utilize the bioS multi5+permute dataset from Allen-Zhu & Li (2023a;b) as our bioS dataset, which generates 5 biography entries per person using randomly chosen sentence templates and permutations. We use the bioR multi5 dataset from them as our bioR dataset, which generates 5 biography entries per person by invoking Llama five times.

Following Allen-Zhu & Li (2023b), we employ GPT2-small (12 layers, 12 heads, and 768 dimensions) for the bioS dataset and GPT2 with 12 layers, 20 heads, and 1280 dimensions

| Query | "The Star-Spangled Banner" is the → | ← national anthem of the United States written by Francis Scott Key. |
|---|---|---|
| **standard** | → national anthem of the United States. | N/A |
| **reverse (*token*)** | → national anthem of the United States. | The Star Spangled Banner is the ← |
| **reverse (*entity**)** | → national anthem of the United States. | The Star-Spangled Banner is the ← |

| Query | Here is how to wash a car: → | ← Those are the steps for washing a car. |
|---|---|---|
| **standard** | → 1. Wash the car with a hose or spray bottle, using warm water and mild soap (such as Dawn dishwashing liquid). Be sure to rinse thoroughly after each wash! 2. Use a soft brush on rough surfaces like windowsills; don't use too much pressure because this can damage paintwork further down below where it is less visible but still important for keeping clean looking cars in good condition over time!. 3 . If needed apply ... | N/A |
| **reverse (*token*)** | → 1. Wash the car with a hose and water, or use a pressure washer to remove dirt from the surface of your vehicle's paintwork. Be sure not to damage any parts that are painted on the body of your car! 2. Rinse off all excess moisture using a sponge or cloth soaked in warm water; this will help prevent rusting over time.3. Wipe down any areas where you have applied wax before moving onto other surfaces such as wheels & tires if applicable (if necessary). ... | ... Step #4 - Clean the surrounding area thoroughly before washing the car. You're done! You are ready to wash the car. Note: Do not leave the car's engine running as soon as you walk away from the car. Leave the car outside for some time to cool down (1-2 minutes), then rinse it out with fresh water. Then pour the clean water back into the car and let it sit in the car until it is completely dry (5-10 minutes, depending on your car). Finally, use a garden hose to get any excess water off of the car. ← |

Table 8: **An example of reversed generations produced by 1.4B pre-trained models:** the model is asked to generate a text completion in both normal (*left*) and reversed (*right*) directions.

| Pre-training method | Finetuning method | NameToDescription | | DescriptionToName | |
|---|---|---|---|---|---|
| | | **forward** | **reverse** | **forward** | **reverse** |
| *Model size: 1.4B* | | | | | |
| reverse (*token*) | reverse (*token*) | 78.3 | 0.0 | 100 | 2.7 |
| reverse (*entity**) | standard | 78.0 | 0.0 | 96.3 | 0.3 |
| reverse (*entity**) | reverse (*word*) | 71.0 | 2.7 | 94.7 | 2.0 |
| reverse (*entity**) | reverse (*entity*) | 77.0 | 78.3 | 95.3 | 2.3 |
| std. (compute-matched) | reverse (*rand k=5*) | 77.0 | 52.3 | 96.0 | 10.7 |
| std. (compute-matched) | reverse (*rand k=10*) | 74.7 | 85.3 | 93.7 | 33.7 |
| std. (compute-matched) | reverse (*rand k=25*) | 77.3 | 96.3 | 97.7 | 70.7 |
| std. (compute-matched) | reverse (*rand k=50*) | 77.3 | 89.3 | 93.0 | 67.3 |

Table 9: Test accuracy (%) on the *fictitious celebrities* task for various different pre-training and finetuning ablation methods.

for the bioR dataset. We also utilize the same AdamW optimizer with cosine learning rate decay ($\beta_1 = 0.9, \beta_2 = 0.98, \varepsilon = 10^{-6}$).

| | FT bdate | FT bcity | FT univ | FT major | FT cname | FT ccity | FT all_to_full | MIX all_to_full | FT four_to_full | MIX four_to_full | FT three_to_full | MIX three_to_full | FT bdate_to_full | MIX bdate_to_full | FT all_to_last | MIX all_to_last | FT four_to_last | MIX four_to_last | FT three_to_last | MIX three_to_last | FT bdate_to_last | MIX bdate_to_last |
|---|---|---|---|---|---|---|---|---|---|---|---|---|---|---|---|---|---|---|---|---|---|---|
| baseline | 0.0 | 0.5 | 0.3 | 1.0 | 0.4 | 13.7 | 0.0 | 0.0 | 0.0 | 0.0 | 0.0 | 0.0 | 0.0 | 0.0 | 0.2 | 0.2 | 0.2 | 0.2 | 0.2 | 0.2 | 0.2 | 0.2 |
| bioS | 100 | 100 | 100 | 100 | 99.9 | 99.6 | 0.0 | 0.0 | 0.0 | 0.0 | 0.0 | 0.0 | 0.0 | 0.0 | 0.1 | 0.2 | 0.2 | 0.1 | 0.2 | 0.2 | 0.2 | 0.1 |
| bioS (token reversal) | 100 | 100 | 100 | 100 | 99.6 | 99.1 | 0.0 | 0.0 | 0.0 | 0.0 | 0.0 | 0.0 | 0.0 | 0.0 | 37.5 | 63.7 | 29.5 | 62.8 | 6.6 | 48.1 | 0.2 | 0.2 |
| bioS (word reversal) | 100 | 100 | 100 | 100 | 99.7 | 99.2 | 0.0 | 0.0 | 0.0 | 0.0 | 0.0 | 0.0 | 0.0 | 0.0 | 84.2 | 99.3 | 77.8 | 99.0 | 19.3 | 91.1 | 0.2 | 0.1 |
| bioS (entity reversal) | 100 | 100 | 100 | 100 | 99.8 | 99.5 | 83.9 | 99.0 | 74.9 | 98.8 | 23.9 | 87.8 | 0.0 | 0.0 | 0.2 | 0.3 | 0.2 | 0.3 | 0.2 | 0.3 | 0.2 | 0.2 |
| bioS (random k=25) | 97.2 | 100 | 100 | 100 | 99.7 | 99.2 | 95.7 | 99.8 | 89.7 | 98.5 | 37.6 | 80.7 | 0.0 | 0.0 | 1.0 | 1.1 | 0.9 | 1.0 | 0.6 | 0.6 | 0.2 | 0.2 |
| bioR | 99.9 | 99.8 | 99.9 | 99.7 | 99.8 | 92.1 | 0.0 | 0.0 | 0.0 | 0.0 | 0.0 | 0.0 | 0.0 | 0.0 | 0.2 | 0.2 | 0.2 | 0.1 | 0.2 | 0.1 | 0.2 | 0.1 |
| bioR (token reversal) | 99.9 | 99.7 | 99.9 | 99.7 | 99.8 | 91.0 | 0.0 | 0.0 | 0.0 | 0.0 | 0.0 | 0.0 | 0.0 | 0.0 | 53.6 | 60.9 | 25.7 | 58.5 | 9.6 | 49.1 | 0.3 | 0.2 |
| bioR (word reversal) | 99.9 | 99.7 | 99.9 | 99.8 | 99.8 | 90.1 | 0.0 | 0.1 | 0.0 | 0.0 | 0.0 | 0.0 | 0.0 | 0.0 | 97.4 | 99.2 | 63.1 | 98.4 | 24.3 | 94.4 | 0.3 | 0.2 |
| bioR (entity reversal) | 99.9 | 99.7 | 99.9 | 99.7 | 99.8 | 90.0 | 96.4 | 97.8 | 53.0 | 92.2 | 23.2 | 78.7 | 0.0 | 0.1 | 0.3 | 0.4 | 0.3 | 0.4 | 0.3 | 0.3 | 0.2 | 0.2 |
| bioR (random k=25) | 99.6 | 98.6 | 99.6 | 99.7 | 99.6 | 89.7 | 98.6 | 98.9 | 95.4 | 97.8 | 84.2 | 94.9 | 0.0 | 0.1 | 12.6 | 8.6 | 9.8 | 8.4 | 8.2 | 7.0 | 0.2 | 0.2 |

Table 10: Forward vs. reversal task accuracy for the data bioS, bioR (Allen-Zhu & Li, 2023b).

**Left block** = forward QA accuracy (ask for fields given person names).
**Middle block** = reversal QA accuracy (ask for person's full name given selected bio fields):
"bdate" = given birthdates only, "all" = given all fields, etc, see Appendix C).
**Right block** = reversal QA accuracy (ask for last name given selected biography fields).
**FT** = pre-train followed by instruction finetune; **MIX** = add instruction FT data to pre-train.

- For the bioS dataset, we pretrain / mix-train for 80,000 steps with a batch size of 192, which is twice their batch size.
- For the bioR dataset, we pretrain / mix-train for 150,000 steps with a batch size of 192, which is twice their batch size.

During pre-training (or mixed-training), we use a weight decay of 0.03 and select the best among three learning rates: 0.0005, 0.001, 0.002; we also employ 1000 steps of learning rate warmup. During finetuning (FT), we use a weight decay of 0.01, and select the best among two learning rates: 0.0003 or 0.0005; we do not use learning rate warmup. During mixed-training, we use $QA_r = 0.3$ which means 30% of the training tokens come from instruction finetune data.

**Reversal QA tasks.** We consider four reversal tasks from Allen-Zhu & Li (2023b):

- Give me the [last/full] name of the person born on October 2, 1996? (bdate_to_last, bdate_to_full)
- Give me the [last/full] name of the person who studied Communications at Massachusetts Institute of Technology and worked for Meta Platforms? (three_to_last, three_to_full)
- Give me the [last/full] name of the person who studied Communications at Massachusetts Institute of Technology, was born in Princeton, NJ, and worked for Meta Platforms? (four_to_last, four_to_full)
- Give me the [last/full] name of the person who studied Communications at Massachusetts Institute of Technology, was born on October 2, 1996 in Princeton, NJ, and worked for Meta Platforms at Menlo Park, CA? (all_to_last, all_to_full)

**Forward QA tasks.** We consider the same six forward tasks from Allen-Zhu & Li (2023a):

- What is the birth date of Anya Briar Forger?
Answer: October 2, 1996.
- What is the birth city of Anya Briar Forger?
Answer: Princeton, NJ.
- Which university did Anya Briar Forger study?
Answer: Massachusetts Institute of Technology.
- What major did Anya Briar Forger study?
Answer: Communications.
- Which company did Anya Briar Forger work for?
Answer: Meta Platforms.
- Where did Anya Briar Forger work?
Answer: Menlo Park, CA.

Full results are summarised in Table 10. One may notice that our reported forward task accuracies are slightly higher than those reported in Allen-Zhu & Li (2023a). This improvement is attributed to our use of a larger batch size, smaller weight decay, beam size 1 without sampling (instead of 4 in their paper), and the best result among three runs.

# D Generation examples

In this section, we show some examples of generation produced by the corresponding models.

**Real-world celebrity task**   Table 11 provides examples of models' generations in response to simple questions about celebrity-parent relationships. All models have 1.4B parameters and were only pretrained.

| Pre-training method | Generation |
|---|---|
| *direct question: "The father of Emma Roberts is"* | |
| std. (compute-matched) | Dylan O'Brien |
| reverse (token) | Paul Robert Thomas |
| reverse (*entity**) | Eric Roberts |
| *direct question: "The child of Gustav Schwarzenegger is"* | |
| std. (compute-matched) | Christina Schwarzenegger |
| reverse (token) | Gustav Schwarzenegger |
| reverse (*entity**) | Arnold Schwarzenegger |

Table 11: Generations produced by models in response to a corresponding prompt from the *real-world celebrity* task. Correct output is colored in green, incorrect generation is red.

**Fictitious facts task**   In Table 12 we show a few examples of models' generations in response to the prompts from a *fictitious celebrity* task. All models are initialized from the fully pre-trained 1.4B model, and further finetuned as described in Section 3.4.

| Finetuning method | Generation |
|---|---|
| *Name2Description, reverse question:* | *"Immersed in the world of directing the virtual reality masterpiece, "A Journey Through Time""* |
| std. (compute-matched) | Sariya Breckenridge has become a household name. |
| reverse (token) | Nola Westbrook remains grounded. |
| reverse (*entity**) | Daphne Barrington has become a household name. |
| *Description2Name, reverse question:* | *"In the realm of Zachary Norwood, a figure who is celebrated for "* |
| std. (compute-matched) | being the first person to complete a solo voyage around the world in a submarine. |
| reverse (token) | saving countless lives during the Great Earthquake of 2065 |
| reverse (*entity**) | saving countless lives during the Great Earthquake of 2065 |

Table 12: Generations produced by models in response to a corresponding prompt from the *fictitious facts* task. Correct output is colored in green, incorrect generation is red. Questions are asked in reverse order with respect to how model has seen these facts during finetuning.

