# OpenReview forum: "Reverse Training to Nurse the Reversal Curse"
_colmweb.org/COLM/2024/Conference — COLM_

### Official Review · Reviewer_nRfh · 2024-05-10

**Rating:** 6
**Confidence:** 4
**Ethics Flag:** 1

**Summary:**

The paper present the problem of reversal curse and a simple approach to help alleviate the issue. The reversal curse is the phenomenon that LLMs seem to lack the capability to properly understand relations such as "Paris is the capital of France", demonstrated by the models' inability to infer from the previous that "The capital of France is Paris". The proposal made in the paper is to pre-train or fine-tune LLMs with data containing reversed text (not just standard left-to-right). The paper investigates multiple approaches to revering text and evaluates model performance in both reversal comprehension and standard LLM evaluation benchmarks. The finding is that models trained with reversed text perform better in the reversal tasks and the performance on standard evaluation benchmarks does not deteriorate, indicating that the approach would be a suitable method for alleviating the reversal curse. The paper is well written, well structured and contains good selection of evaluation results. The only bigger issue with the paper is the lack of qualitative evaluation of the model outputs. It is well known that many of the evaluation benchmarks and tasks can be solved by models that do not really "understand" the language or the task or by models that otherwise generate poor quality language. Including qualitative evaluation and example model outputs would help the reader understand what effect the reverse training has on the model.

**Questions To Authors:**

I would like to see more qualitative evaluation of model output, comparing baseline and revere trained versions.

**Reasons To Accept:**

The paper presents a simple but effective data augmentation method that can help alleviate the reversal curse in LLMs. The paper is well written, the main arguments are well presented and the experimental results support the arguments.

**Reasons To Reject:**

Intuitively, training on reversed text should harm the model in terms of how natural the output of the model is. In addition to simple benchmark test results, I would have liked to see some more qualitative evaluation showing what the impact of reverse training has on the model output.

---

> ### Author Rebuttal · Authors · 2024-05-30
>
> **Intuitively, training on reversed text should harm the model in terms of how natural the output of the model is**
>
> That’s a valid question but if the reversal data looks significantly different from the original data, the language model can learn an internal “switch” and when prompted with normal English, it is going to continue with normal English without reversal. This has been systematically studied for instance in https://arxiv.org/abs/2305.13673, where the authors show that if you train a transformer with both grammatically correct AND incorrect data, then if you prompt the model with the grammatically correct one, it will not continue the generation with grammar mistakes (grammar mistakes only happen if (1) you prompt with grammar mistake data and (2) you increase the generation temperature).
>
> In a structural sense, the original and reversal data have significantly different syntax, even if they have the same words, so that the model can learn to distinguish them easily. In that sense they are closer to two different languages, and again language models have no problem learning multiple natural languages at once, and using the right one at the right time.
>
> **In addition to simple benchmark test results, I would have liked to see some more qualitative evaluation showing what the impact of reverse training has on the model output**
>
> Thank you for your feedback. We already included Table 8 with examples of reverse generations. Below we show a sample of long-form generation from our 1.4B reverse pre-trained model. As you can see, the reverse-trained model is still able to generate coherent text in the forward direction, and it is not qualitatively different from the baseline model.
>
> | _Prompt_: “Here is how to wash a car:” |
> |:-------------|
> | _**Generation:**_ “1. Wash the car with a hose and water, or use a pressure washer to remove dirt from the surface of your vehicle’s paintwork. Be sure not to damage any parts that are painted on the body of your car! 2. Rinse off all excess moisture using a sponge or cloth soaked in warm water; this will help prevent rusting over time.3. Wipe down any areas...”
>
> We will add full examples in the final version. In addition, we will also add generation examples from the celebrity task, which appear to work fine too (which correlates with the metrics in the paper).
>
> We are unable to add more generations due to the space limit, but as we understand we could reply with more information in the discussion phase.

---

> > ### Comment · Reviewer_nRfh · 2024-06-05
> >
> > Thanks for the andditional information.

---

### Official Review · Reviewer_8jTM · 2024-05-11

**Rating:** 7
**Confidence:** 4
**Ethics Flag:** 1

**Summary:**

Motivation of the paper is that language models do not learn the reverse of facts (e.g., "X has feature Y" versus "Y is a feature of X") due to their left-to-right training and the pre-training data including facts in only one of those two ways. This causes models to not be able to answer some questions based on how it's asked (e.g., "What is a feature of X?" versus "What is Y a feature of"), even though the fact is in its training dataset.

- The symbolic reverse task allows a controlled study of the approach and provides clear validation of the authors' idea.
- The "celebrity task" has very low accuracy numbers, seemingly too low to make any conclusions. The data appears to be simply insufficient for the model to perform this task.
- Real-world larger-scale tasks show that using the reverse training approach during pre-training and/or fine-tuning can be helpful. When the same amount of training data is used (data-matched) the proposed approach is able to solve previously unsolvable tasks, with pretty good accuracy numbers. The best approach doesn't seem to require the reverse training during pre-training but just during fine-tuning.

**Questions To Authors:**

- Do you think this would work for other languages? Given multilingual nature of LLMs, some discussion of this would be good.
- typo in Section 4 "Mitgiations"
- I wonder if the "reversal curse" is just a symptom of a deeper issue with language models' logical reasoning abilities. While LLMs are known to reason over the facts they've been trained on, this could be just one of many reasoning tasks they are bad at. I would encourage the authors to provide their thoughts on this.

**Reasons To Accept:**

The authors present a very neat idea that has merit and can be very impactful as it is easy to apply while training language models. Their presentation is very clear with extra details provided in the Appendix. Results are pretty impressive given the simplicity of the approach.

**Reasons To Reject:**

A direct empirical comparison with Guo et al 2024 is needed to determine the value of this approach. While it is a more complex augmentation and therefore can only be applied at fine-tuning stage, I am not convinced by the mixed results that doing the augmentation during pre-training will provide the best results in a real-world scenario. Tables 6 and 7 imply that the best approach would be to do standard pre-training and reverse fine-tuning. In that case, I would've liked to see a comparison with Guo et al's work where both approaches are applied during fine-tuning only.

---

> ### Author Rebuttal · Authors · 2024-05-30
>
> **A direct comparison with Guo et al 2024 …**
>
> This work is concurrent (same month as the submission date), thus direct comparison was not possible.
>
> **the best approach would be to do standard pre-training and reverse fine-tune …**
>
> Importantly, this is only true if we are testing on facts that are learned during finetuning, which is true for Table 6 – these fictitious celebrities are seen in finetuning, but not in pretraining. However, in practice, most of the knowledge LLMs have is obtained during the pre-training phase, thus reverse fine-tuning will not be sufficient in solving the reversal curse. A similar observation is made in Allen-Zhu & Li (2023b). Hence we recommend reverse pretraining.
> We have now also performed additional experiments by training a 1.4B model from scratch using the “random reversal” method, and further finetuned on the fictitious and real celebrity tasks, which was missing before. Evaluations outperform all other models, and show the importance of learning the “reversing” skill during finetuning. We will update Tables 5-7 accordingly. E.g. Table 6 would have additionally a 1.4B Model:
>
> | Pre-train Method   | Finetune Method | Name2Desc (fwd/rev) | Desc2Name  (fwd/rev )|
> |:-------------|:--------------:|--------------:|--------------:|
> | reverse (rand k=25)        | reverse (rand k=25)          | 81.0*   /    97.0*      |   97.3   /   73.0 |
>
> We have more results but cannot put here due to limits to reply length.
>
> **Table 5 low accuracy numbers**
>
> The numbers are low, but there are clear differences between the models. Our reasoning is: (1) the model is only 1.4B parameters, this improves with size; (2) there is no fine-tuning, only pre-training for the task. The task is realistic as we know SOTA LLMs can easily recall (forward, but not reverse) facts about real celebrities.
>
> **Do you think this would work for other languages**
>
> We believe our method will generalize to other languages well, specially the random-segment reversal version because that will have less dependence on language. We will add this discussion to the paper.
>
> **reversal curse is a symptom of a deeper issue with LLMs' reasoning abilities**
>
> The reversal curse may have multiple underlying causes. The one-directional training objective is causing this as we demonstrated in our paper. There could be an architectural issue on how facts are being stored in the model. But even such an (unknown) architectural fix might benefit from reverse training.

---

> > ### Comment · Reviewer_8jTM · 2024-06-04
> > **Generalizing to other languages**
> >
> > Thank you for the clarifications. I do want to dig into the multilingual aspect a bit more, as I think that the merit of this approach is
> >  mainly for practitioners (as you also stated in your rebuttal), and that depends on how well the method generalizes beyond English.
> >
> > In your rebuttal to my fellow reviewer, you said "In a structural sense, the original and reversal data have significantly different syntax, even if they have the same words, so that the model can learn to distinguish them easily."
> >
> > What if the language has free word order? Would that not violate your assumption that the original and reversal data have significantly different syntax, therefore the model would be able to implicitly learn a "switch"?

---

> > > ### Author Response · Authors · 2024-06-04
> > >
> > > Thank you for the response. We agree that the method should work with other languages to be practical. We assumed that most languages do have certain word ordering, so reverse text would be distinguishable by the model. However, if that’s impossible because the language has free word order (in the extreme case where reverse text is not distinguishable both syntactically and semantically, then the model might not need to switch), all we need to do is add a special token at the beginning of each reverse sequence fed to the model indicating that the sequence is in reverse order (normal sequences will have normal BOS token but no special token). This will help the model distinguish reverse sequences from normal ones at the very first token. Actually, we already do this in random-segment reversal with “[REV]” token (see Tab 2) because random-segment reversal is hard to distinguish from the normal sequences (we also use this token in the middle to indicate segment boundaries). Similarly, we can insert “[REV]” to the beginning of reversed sequences for the other types of reversals (token, word, entity-preserving) during training so the model will not be confused. We will add this discussion to the paper.

---

### Official Review · Reviewer_SEXk · 2024-05-11

**Rating:** 6
**Confidence:** 3
**Ethics Flag:** 1

**Summary:**

This paper proposes to use reverse training to mitigate the reversal curse in language models.
In short, language models have been shown by prior work to present a "reversal curse": models are typically presented with strings of the form, e.g. "Paris is the capital of France", and they struggle to answer questions such as "what is the capital of France?"; this is because they are rarely presented with strings where "Paris" follows "France" (as in, e.g., "in France's capital, Paris, ...").
The authors then show that, when language models are trained on a mix of forward and reverse strings (using a standard language modeling next token prediction objective), LMs perform much better on questions which require such a "reverse answer".

**Reasons To Accept:**

The paper investigates an interesting topic: why do language models perform better on facts written with a certain directionality than another.

This paper also performs a large number of analyses related to the impact of reverse training on model performance. It studies the impact of reverse training in: a toy task, pretraining, and fine-tuning. Further, it considers the impact of reversing strings at the token, word, or entity level. Finally, it studies the impact of reverse training not only on tasks in which the model originally performs poorly, but also investigates whether reverse training hurts model performance on tasks the model originally did well in.

**Reasons To Reject:**

While reverse training seems to experimentally help with the reversal curse issue, it seems to focus on solving a "symptom" caused by how language models are trained/defined rather than trying to fix the source of the issue. Identifying the source of why language models are inconsistent in representing facts based on the directionality in which they are presented, and proposing a principled way to fix it, would make this paper's contribution stronger in my opinion.

---

> ### Author Rebuttal · Authors · 2024-05-30
>
> Thank you for your thoughtful review.
>
> We think that the training "direction" (right-to-left vs. left-to-right) is an important aspect of the reversal curse, and it may underlie one of the core issues. Only predicting the next token is an artificial constraint and may not be the only way to build capable models. Our argument is that this one-directional objective is one of the reasons why language models suffer the reversal curse. While we agree that there can be other architectural reasons that contribute to the issue, it might be challenging to solve the reversal curse without modifying the next-token prediction objective, as shown in Allen-Zhu & Li (2023b). From that paper: we know that (1) bidirectional models such as BERT with MaskedLM don’t solve the problem, (2) adding abundant reversal finetune data doesn’t work; (3) pushing reversal finetune data to the pretrain stage doesn’t solve the problem. Recent theoretical work such as https://arxiv.org/abs/2405.04669 also suggests fundamental limits to the current auto-regressive models due to asymmetry. Fortunately, we feel that if there’s a simple way to fix the issue on the data level, this could be beneficial to practitioners.

---

> > ### Comment · Reviewer_SEXk · 2024-06-07
> > **Response to Authors**
> >
> > I thank the authors for their response. I've read both the other reviews, and the authors' responses, and I am keeping my scores.

---

### Decision · Program_Chairs · 2024-07-10

**Decision:**

Accept

**Comment:**

This paper proposes reverse training for solving the reverse curse. The method is simple but effective. All reviews are positive.